# ONX 0914 Lacks Selectivity for the Cardiac Immunoproteasome in CoxsackievirusB3 Myocarditis of NMRI Mice and Promotes Virus-Mediated Tissue Damage

**DOI:** 10.3390/cells9051093

**Published:** 2020-04-28

**Authors:** Hannah Louise Neumaier, Shelly Harel, Karin Klingel, Ziya Kaya, Arnd Heuser, Meike Kespohl, Antje Beling

**Affiliations:** 1Charité—Universitätsmedizin Berlin, Corporate Member of Freie Universität Berlin, Humboldt-Universität zu Berlin, and Berlin Institute of Health (BIH), Institute of Biochemistry, 10117 Berlin, Germany; hannah-louise.neumaier@charite.de (H.L.N.); shellyharel@web.de (S.H.); meike.kespohl@charite.de (M.K.); 2Institute for Cardiopathology, University of Tuebingen, 72074 Tuebingen, Germany; karin.klingel@med.uni-tuebingen.de; 3Medizinische Klinik für Innere Medizin III: Kardiologie, Angiologie und Pneumologie, Universitätsklinikum Heidelberg, Medizinische Klinik für Innere Medizin III: Kardiologie, Angiologie und Pneumologie, Universitätsklinikum Heidelberg, 69120 Heidelberg, Germany; ziya.kaya@med.uni-heidelberg.de; 4Deutsches Zentrum für Herz-Kreislauf-Forschung (DZHK), partner side Heidelberg/Mannheim, 69120 Heidelberg, Germany; 5Max-Delbrueck-Center for Molecular Medicine, 10115 Berlin, Germany; heuser@mdc-berlin.de; 6Deutsches Zentrum für Herz-Kreislauf-Forschung (DZHK), partner side Berlin, 10785 Berlin, Germany

**Keywords:** myocarditis, proteasome, immunoproteasome inhibitor, inflammation, proteostasis

## Abstract

Inhibition of proteasome function by small molecules is highly efficacious in cancer treatment. Other than non-selective proteasome inhibitors, immunoproteasome-specific inhibitors allow for specific targeting of the proteasome in immune cells and the profound anti-inflammatory potential of such compounds revealed implications for inflammatory scenarios. For pathogen-triggered inflammation, however, the efficacy of immunoproteasome inhibitors is controversial. In this study, we investigated how ONX 0914, an immunoproteasome-selective inhibitor, influences CoxsackievirusB3 infection in NMRI mice, resulting in the development of acute and chronic myocarditis, which is accompanied by formation of the immunoproteasome in heart tissue. In groups in which ONX 0914 treatment was initiated once viral cytotoxicity had emerged in the heart, ONX 0914 had no anti-inflammatory effect in the acute or chronic stages. ONX 0914 treatment initiated prior to infection, however, increased viral cytotoxicity in cardiomyocytes, promoting infiltration of myeloid immune cells into the heart. At this stage, ONX 0914 completely inhibited the β5 subunit of the standard cardiac proteasome and less efficiently blocked its immunoproteasome counterpart LMP7. In conclusion, ONX 0914 unselectively perturbs cardiac proteasome function in viral myocarditis of NMRI mice, reduces the capacity of the host to control the viral burden and promotes cardiac inflammation.

## 1. Introduction

The ubiquitin-proteasome system (UPS) is an integral part of cellular proteostasis, functioning as the primary route for intracellular degradation of misfolded, damaged, or short-lived proteins [1]. The ubiquitin conjugation machinery marks these proteins for recognition by the proteolytic complex, the proteasome. The proteasome is a barrel-shaped, twofold symmetric, multi-subunit enzyme that is flanked by regulatory 19S regulatory particles (19S), which serve to bind protein substrates. Peptide hydrolysis is exclusively carried out by β1, β2, and β5 subunits within the inner chamber of the 20S core complex. Particularly under conditions of cellular stress, the UPS needs to adapt its protein turnover capacity to ensure elimination of the increasing abundance of unfolded and potentially toxic proteins [2]. In response to inflammatory cytokines [3,4], DNA-damage [5], or oxidative stress [6], cells selectively upregulate the expression of the immunoproteasome (i-proteasome), a specific proteasome isoform, containing alternative catalytic subunits (β1i/LMP2, β2i/MECL-1, β5i/LMP7) [7]. In comparison to its standard proteasome counterpart, the i-proteasome accelerates the proteolysis of specific peptide substrates [8,9] and allows for the facilitated degradation of oxidant-damaged proteins, which may accumulate during inflammation, for example [3,10,11]. While this specific adaptation of proteasome function is of particular relevance in non-immune cells with high standard proteasome abundance under basal conditions, immune cells exhibit high i-proteasome and low standard proteasome expression levels in their naive state. This cell-specific proteasome-isoform expression pattern in non-immune and immune cells initiated the structure-guided design of specific proteasome inhibitors, specifically targeting the i-proteasome [12,13].

Based on the IFN-γ-dependent upregulation of the i-proteasome subunits, which are in close proximity to the genes of the MHC II region, and their unique cleavage properties, the i-proteasome was initially seen as a prerequisite needed for efficient MHC class I antigen presentation [4,14]. Undisputedly, the i-proteasome strongly enhances CD8^+^ T cell responses for some pathogens [15,16,17], yet there are also examples where the standard proteasome can efficiently compensate for the lack of intact i-proteasome activity [18,19]. Of the different compounds which can experimentally accomplish inhibition of the i-proteasome, ONX 0914 is the best characterized [20,21]. Whereas physiologically desired effects achieved by treatment with the epoxyketone ONX 0914 requires a combined inhibition of both LMP7 and LMP2 [20,21], there is also a LMP7-selective dipeptide inhibitor available which shows potent immunosuppressive activity [22]. Another compound referred to as PRN1126, developed as an exclusively LMP7-specific inhibitor shows, however, limited in vivo effects on adverse immune responses [21]. Nevertheless, the availability of selective i-proteasome inhibitors paved the way towards the definition of new functions of the i-proteasome. These involve control of DAMP- or PAMP-triggered signaling responses, resulting in elevated production of cytokines, e.g., by innate myeloid cells, altered T cell activation and differentiation, B cell function, or immune cell survival [23,24,25]. Based on these multidimensional immune cellular functions, it was expected that i-proteasome inhibitors would be capable of hindering inflammation-driven carcinogenesis [26,27], autoimmune-related inflammation [20,28,29], or transplant rejection [22,30].

Inhibiting the i-proteasome, however, in response to infection-triggered inflammation is a double-edged sword, since the repertoire of cellular processes affected by the i-proteasome ranges from proper immune cell activation needed for pathogen control to potentially harmful events, known as immunopathology. Particularly for intracellular pathogens, such as fungi, protozoa, viruses, or intracellular bacteria, which rely on successful MHC class I antigen presentation machinery for elimination of infected cells by CD8^+^ T lymphocytes, this limits the general applicability of i-proteasome inhibitors for the treatment of pathogen-triggered inflammation [31,32]. Myocarditis, an inflammatory disease of the heart, is a classic example where both direct pathogen-mediated cellular injury, as well as the resulting activation of an immune response, contribute to acute organ failure and simultaneously predispose patients to chronic cardiac dysfunction [33]. Since in the Western world, non-ischemic cardiac inflammatory disease is most commonly caused by viral infections, our research group, which is interested in dissecting the pathology behind this disease, relies on the well-established mouse model of Coxsackievirus B3 (CVB3) myocarditis [33]. Infection with CVB3, a single-stranded RNA virus of the Picornaviridae family, and the immune response that follows, induce a sequence of organ-specific and systemic pathologies. Using the intraperitoneal inoculation route, the virus initially causes a pancreatitis, resulting in destruction of exocrine cells of the pancreas [34], and subsequently myocarditis, showing variable severity in different laboratory mouse strains [35]. Depending on the strain-specific susceptibility to the disease, mice develop a heterogeneous acute [8,36] and sometimes a chronic stage of the disease [37], providing an opportunity for research into the development of chronic sequela, such as inflammatory cardiomyopathy, which can manifest after acute myocarditis. The i-proteasome plays an important, yet not fully explained role in the process of CVB3-triggered cardiac inflammatory immune response, with both knockout and inhibition of the multi-catalytic protease exacerbating the disease in C57BL/6 mice [3,38]. The i-proteasome is strongly upregulated in heart tissue during viral myocarditis and its induction involves type I IFNs [8] as well as IFN-γ [39]. In C57BL/6 mice, the i-proteasome is required for the timely degradation of oxidant-damaged proteins, to prevent their accumulation in inflamed cells and tissues [3,11,40]. Moreover, the i-proteasome regulates the production of innate immune regulators, such as pentraxin3 [41], with failure of such processes contributing to enhanced cardiac tissue damage. In a previous work, however, we demonstrated that ONX 0914 can act in the opposite manner, clearly protecting another laboratory strain, namely A/J mice, from developing CVB3 myocarditis. In this host, i-proteasome activity is detrimental, since the comparatively severe pathology triggered by CVB3 infection in this strain is primarily attributed to an overall adverse immune response, resulting in a sepsis-like cytokine storm and distributive shock condition, all of which can be mitigated by i-proteasome inhibitors [38].

The magnitude of the acute inflammatory injury, triggered in the heart by cardiotropic viruses, predisposes a host to cardiac remodeling processes that involve classical wound healing processes with the differentiation of myofibroblasts and fibrosis formation [33]. Shifting focus from acute towards the chronic stages of the disease, we questioned whether i-proteasome inhibitors could potentially provide a targeted solution for attenuating disease progression to chronic stages—for where there is reduced acute heart muscle damage, subsequent scarring and cardiac remodeling can be expected to be more limited as well [42]. Although the A/J strain has been noted to form chronic inflammatory lesions [43], mice become severely ill during acute myocarditis and can partially succumb to infection [44]. In our hands, A/J mice are not ideally suited to investigate the chronic state. NMRI mice, however, while susceptible to acute CVB3-myocarditis, show much less impact on their general well-being and have consistently been shown to develop chronic viral myocarditis [37,45], making them a favored model for later stages of the disease. Regarding the positive effects, ONX 0914 treatment demonstrated in acute myocarditis in A/J mice, we asked the question whether i-proteasome inhibition by ONX 0914 might attenuate the development and severity of chronic myocarditis in NMRI mice as well.

## 2. Results

### 2.1. Influence of ONX 0914 on Myocarditis in NMRI Mice

Our previous work demonstrated a protective effect of ONX 0914 on acute inflammatory tissue injury of the heart after infection of A/J mice with a cardiotropic CVB3 strain [38]. To investigate how ONX 0914 influences a late consequence of viral infection, we used a well-characterized and frequently used mouse model for chronic viral myocarditis, inoculating outbred NMRI mice with CVB3 strain 31-1-93 [37]. We applied a therapeutic approach, with the initiation of treatment on day 3 post-infection [46], once viral injury of the heart tissue has emerged. Mice were randomized to a control (vehicle-treated) or ONX 0914 (inhibitor-treated) group and received daily doses from day 3 until day 8. Thereafter, until read out on day 28, treatment was continued three times per week (Figure 1A). During chronic infection, animals treated with ONX 0914 showed a significantly slower recovery from the initial weight-loss which mice typically develop during acute viral infection (Figure 1B). Analysis of the virus concentration in infected heart tissue revealed similar viral RNA copy numbers in mice treated with ONX 0914 (Figure 1C), with no detection of infectious viral particles in either vehicle- or inhibitor-treated groups on day 28. Troponin T (TnT) levels, measured using serum drawn on day 28, showed values equal to those of uninfected controls, demonstrating that there is no ongoing myocardial tissue destruction at this stage of infection in ONX 0914-treated groups (data not shown).

Echocardiography was performed to determine if there is any impact on cardiac function by ONX 0914 treatment (Table 1). Parameters that describe cardiac performance, such as cardiac output and ejection fraction of the left ventricle, were within the range of those detected in naive mice prior to infection, revealing no measurable influence due to ONX 0914 treatment. Altogether, these data provide evidence that post-viral cardiac remodeling, regardless of ONX 0914 treatment, has no biologically relevant effect on cardiac function in NMRI mice. Although the overall architecture of the heart muscle was not heavily affected, cardiac sections scored for cardiac remodeling processes, comprising post-viral inflammatory damage and formation of fibrosis, revealed signs of tissue injury (Figure 1D–F). Similar to the vehicle-treated group, ONX 0914 treatment led to elevated scores on day 28, clearly arguing against a protective effect of ONX 0914 on post-viral heart tissue injury in NMRI mice.

Since we found no functional deterioration at the chronic stage in infected NMRI mice, we asked how a therapeutic administration of ONX 0914 affects the virus-triggered inflammatory tissue damage during acute viral myocarditis, peaking at around day 8 post-infection, using NMRI mice. Similar to the first experiment, mice received ONX 0914 or the vehicle daily from day 3 after infection until day 8 (Figure 2A). During the course of acute infection up to day 8, ONX 0914 had no effect on body weight (Figure 2B). In ONX 0914-treated mice, we found a trend towards a slightly increased CVB3 concentration on day 8 post-infection (Figure 2C). Histological scoring of heart tissue, comprising myocardial necrosis and inflammation at this stage, revealed similar scores in both groups (Figure 2D/E). Overall, we found that ONX 0914 with treatment initiated after 3 days had no protective effects on acute viral myocarditis in NMRI mice. In fact, there was a tendency towards mild exacerbation of cardiac inflammation in the ONX 0914 group, which led us to ask whether the i-proteasome might, as shown for C57BL/6 mice [3,38,41], play a beneficial role during CVB3 myocarditis in NMRI mice.

To investigate the function of the i-proteasome in more depth, either ONX 0914 or vehicle was administered to mice daily for 8 days, starting one day prior to infection (Figure 3A), which ensured inhibition of the i-proteasome during the entire course of infection [38]. On day 0, mice were infected with CVB3. Regarding their overall body weight, there were no significant differences in the two treatment groups, with all mice experiencing weight loss, as is known for infection with CVB3 (Figure 3B). Measurement of serum TnT concentrations at day 8, however, revealed elevated TnT levels in inhibitor-treated animals in comparison to their controls, demonstrating that the myocardial injury was even more pronounced in those animals that were treated with ONX 0914 (Figure 3C). Correspondingly, histological scoring of HE-stained cardiac sections showed myocardial injury in ONX 0914-treated animals was, if anything, more severe (Figure 3D,E). Quantitative analysis of infiltrating immune cells, performed by flow cytometry of cardiac tissue, revealed a higher number of inflammatory monocytes (Ly6C^high^) in hearts obtained from ONX 0914-treated mice (Figure 3F). In viral myocarditis, the virus concentration and the cytotoxicity it induces is a main determinant for the severity of the inflammatory response. Both the evidence for greater myocardial tissue injury and the elevated number of myeloid immune cells in the heart tissue of ONX 0914-treated NMRI mice suggest that ONX 0914 influences the virus concentration in this host. In fact, we found that the viral load, as determined by quantitative PCR and plaque assay, was increased in mice treated with ONX 0914 in comparison to their controls (Figure 3H,I). Viral RNA is a classical trigger of IFN and cytokine production and we thus investigated how ONX 0914 influences the expression of these molecules in infected heart tissue. In ONX 0914-treated mice, we found an elevated IFN-β production, yet an unaffected IFN-stimulated gene signature as representatively shown by IFIT1 and IFIT3 expression. With the exception of CXCL-10, where ONX 0914 treatment resulted in higher expression levels, the mRNA expression of pro-inflammatory and chemoattractant cytokines was not affected by ONX 0914 treatment (Figure 3J).

To investigate how ONX 0914 treatment in acute myocarditis influences ventricular filling, vascular tone, and resultant cardiac performance, we performed echocardiography (Table 2).

At the acute state of myocarditis, we found a reduced cardiac output in both treatment groups. Nevertheless, a closer investigation of the underlying aspects that might contribute to this lower cardiac output in infection illustrated different findings in the vehicle- and ONX 0914 groups. In the vehicle group, we observed a decrease of the left ventricular filling, which is indicative for a distributive failure of the vascular tone. Since the end-diastolic volume, reflecting left ventricular filling, was not altered by CVB3 infection in the ONX 0914 group, the treatment of NMRI mice with ONX 0914 apparently compensates for this vascular failure. On the other hand, we found a reduced left ventricular ejection fraction in this group. In line with the elevated myocardial cytotoxicity found in ONX 0914-treated mice, this might suggest systolic dysfunction as a potential effector of the reduced stroke volume. In vehicle-treated NMRI mice, we found no relevant decline of the ejection fraction at the acute state of myocarditis.

### 2.2. Influence of ONX 0914 on the Molecular Architecture of the 20S Proteasome Complex in Viral Myocarditis

Altogether, our data demonstrate that, at the acute stage of inflammation, ONX 0914 treatment promoted viral cytotoxicity, as reflected by cardiac myocyte damage, leading to elevated serum troponin T levels, and resultant inflammatory injury. In contrast to previous reports involving A/J mice [38], ONX 0914 had no relevant effect on systemic signs of viral infection, such as weight loss, cardiac output, or survival. Moreover, we found no evidence of impaired virus control in other tissues, such as spleen and pancreas (data not shown). Based on this controversial development of viral burden in different tissues, we questioned the inhibitory capacity of ONX 0914 with respect to the cardiac i-proteasome. The spleen, which shows no effect of ONX 0914 on the virus concentration (Appendix A, is a secondary lymphatic organ with important immune functions and the amount of i-proteasome expressed in naive spleen far outweighs that of the heart. These aspects make it an interesting organ to investigate the effects promoted by viral infection and ONX 0914 treatment on the catalytic subunits of the proteasome and to compare our results with those of heart tissue.

For both spleen (Figure 4) and heart tissue (Figure 5), representative mouse samples from vehicle- and ONX 0914-treated groups were analyzed separately, comparing uninfected controls (day 0) for each group to the early phase of infection (day 2) as well as acute (day 8) and chronic (day 28) stages of viral myocarditis. For mice treated with the vehicle, in murine splenic tissue, determination of mRNA levels for the catalytic i-proteasome subunits and their respective standard proteasome counterparts showed an increase of all i-proteasome subunits during acute myocarditis and a return to at least baseline values during chronic myocarditis (Figure 4A). Although temporal i-proteasome profiling was similar in ONX 0914-treated mice, the overall fold-induction was lower for all three i-proteasome subunits (Figure 4B). In cardiac tissue, we noted a similar mRNA expression profile for LMP2, LMP7, and MECL-1, yet with clear evidence of a much stronger induction of mRNA expression (Figure 5A,B), which is in line with pronounced inflammatory responses specifically in the heart during viral myocarditis. Throughout infection, mRNA profiles of standard proteasome subunits remained unchanged in both spleen and heart tissue, regardless of the treatment group.

We then performed Western blot analysis of the proteasome subunits. Corresponding to the respective mRNA profiles, in spleen we found slightly elevated expression levels of the i-proteasome subunits, revealing a mild reduction of the low abundance standard proteasome counterpart in vehicle-treated mice at day 8 post-infection (Figure 4C/E). ONX 0914 treatment had no relevant effects on i-proteasome induction and, other than in vehicle-treated animals, led to increased expression of the standard proteasome, particularly at day 8 post-infection (Figure 4D). To provide a direct comparison between treatment groups, a sample from one representative animal per group and day of infection was loaded onto the same gel (Figure 4E). Inhibition of a catalytic subunit by ONX 0914 is reflected by altered electrophoretic mobility due to an increased molecular weight, resulting in slower migration in SDS PAGE. Overall quantification of such alterations of molecular weight due to ONX 0914 treatment revealed an inhibition of all three i-proteasome subunits during the acute stage of infection in the spleen. 

ONX 0914 preferentially bound to LMP7 and MECL-1, revealing comparably lower proportional inhibition of LMP2. Long-term treatment of mice with ONX 0914 for 28 days decreased the proportion of ONX 0914-bound i-proteasome subunits in comparison to earlier stages of infection (Figure 4F), which might indicate a loss of efficacy for ONX 0914 over time.

As suggested by robust induction of i-proteasome mRNAs in heart tissue during acute infection, protein expression profiling confirmed the pronounced upregulation of the i-proteasome, peaking on day 8 at the acute stage in cardiac tissue and remaining elevated during the chronic stage on day 28, regardless of ONX 0914 treatment.

Standard proteasome expression was decreased to a comparable degree on day 8 (Figure 5C,D). Although ONX 0914 bound to all three i-proteasome subunits, showing a similar preference for LMP7 as seen in splenic tissue, we observed a drop in the efficacy of i-proteasome inhibition during the acute stage on day 8, coinciding with a loss of selectivity for the i-proteasome, particularly at this stage (Figure 5D–F). ONX 0914 also bound to the standard proteasome subunit β5, resulting in its partial inhibition prior to infection. Importantly, at the same time as i-proteasome formation and inhibition increased, we found a shift from partially towards completely inhibited β5 subunits on day 8 (Figure 5F).

Altogether, these data demonstrate that ONX 0914 incompletely inhibits the i-proteasome, formed in infected mouse hearts, and simultaneously affects the standard proteasome subunit β5 to a remarkable extent. Our data indicate a modified inhibitory capacity of ONX 0914 for proteasome subunits in non-immune cells, where the i-proteasome is strongly induced after infection and ONX 0914 shows lower selectivity for the i-proteasome.

## 3. Discussion

Based on the anti-inflammatory capacity accomplished by treating mice with i-proteasome-selective inhibitors, in this study we determined how ONX 0914, a broadly studied epoxyketone-based compound with irreversible inhibitory kinetics, affects virus-induced inflammation and remodeling processes, resulting in chronic myocarditis. Using the NMRI mouse model of CVB3 myocarditis, we found that ONX 0914, when given during the entire acute phase after infection, interfered with host processes needed to counteract virus-induced cytotoxicity and thereby mediated tissue injury, all resulting in severe inflammatory responses. Whereas ONX 0914 was specific for the i-proteasome in tissues, such as spleen, that constitutively have a high abundance of this isoform, the proportion of ONX 0914-bound i-proteasome subunits declined in the heart, when this proteasome isoform accumulated. In this phase, ONX 0914 bound completely to the cardiac standard proteasome subunit β5, which harbors the main chymotryptic-like activity of the proteasome.

The genetic background associated with the immunological make up of NMRI mice, although not well-characterized, is thought to predispose these mice to the development of a chronic course of viral myocarditis [37]. Progression towards chronic heart tissue injury in this strain is a consequence of acute myocarditis, which in A/J mice, showing a severe systemic inflammatory response after infection with cardiotropic CVB3 [44], is principally reversible by ONX 0914 [38]. Nevertheless, our data clearly demonstrate that the profound anti-inflammatory effects induced by ONX 0914, illustrated in the context of both viral infection [38,47] and autoimmunity [20,28,29,48], do not primarily affect the pathogenesis induced by viral infection in NMRI mice. In fact, we found that intact proteasome activity enables mice to cope better with the invading pathogen. The findings of this study echo our previous report on ONX 0914-mediated exacerbation of virus-triggered cardiac pathology in C57BL/6 mice [38] and is supported by the detection of a markedly increased fungal burden in a mouse model of *Candida albicans* infection [31]. For both pathogens, C57BL/6 mice were investigated and ONX 0914 inhibited the innate antiviral and antifungal immunity required for efficient control of these pathogens. The elevated pathogen load triggered the resulting recruitment of myeloid immune cells. Although the present study confirms elevated immunopathology in ONX 0914-treated mice to be most likely attributed to a higher concentration of the pathogen, our findings differ regarding the intact activation e.g., of IFN responses in NMRI mice. We have no experimental evidence, other than that previously reported in C57BL/6 mice [38,49], that ONX 0914 affects innate pathways of the IFN response, which could influence replication and spreading of the virus in NMRI mice as well. The IFN signature was similar in both vehicle- and ONX 0914-treated mice. In fact, corresponding to increased viral load at the acute state of myocarditis, IFN production was enhanced in the ONX 0914 group and this might be a result of the higher pathogen burden, a main determinant for the magnitude of such responses.

An alternative aspect that needs to be considered regarding the exacerbation of viral myocarditis in ONX 0914-treated NMRI mice, involves the initially proposed specific role of the i-proteasome in antigen presentation [15]. One could assume that i-proteasome inhibition impairs CD8 T cell immunity and this determines the reported exacerbation in ONX 0914-treated mice. However, other than intracellular pathogens, such as LCMV [50] or *Listeria monocytogenes* [51], CVB3 infection elicits only a weak expansion of CD8^+^ T effector cells [8,52,53]. Moreover, we demonstrated intact effector T and B cell responses, as well as unaffected memory responses, in both ONX 0914-treated and LMP7-deficient C57BL/6 mice after CVB3 infection [3,38]. Altogether, we conclude that alterations of the adaptive immune response, which might be mediated by ONX 0914 [30,54], are not among its main effectors, leading to exacerbated pathology in CVB3-infected NMRI mice. Quite in contrast to its pleiotropic cellular functions in the regulation of inflammatory signaling cascades and in antigen presentation in the host cell, the UPS can also be utilized to control the abundance of viral proteins in infected cells [25]. Direct interactions between viral and host-cell proteins [55] may offer a putative explanation for how inhibition of the proteasome complex, as shown here for ONX 0914 treatment in NMRI mice, might affect the viral load independent of the innate and adaptive immune responses. Molecular aspects, shedding light onto the connection between the expression of viral proteins and their degradation by the proteasome, are known for enterovirus 71 (EV71), another member of the Picornaviridae family. In wild-type EV71, sumoylation, a post-translational ubiquitin-like protein modification, of the viral protease 3C directs this viral protein to the ubiquitination apparatus, marking it for degradation by the proteasome and correlating with lower virus production. Clinically relevant EV71 strains with enhanced virulence, however, lack the sumoylation site of the viral protease 3C, thereby increasing virus protein stability and promoting viral replication [56]. This exemplary mechanistic illustration of viral and host-cell protein interaction elegantly demonstrates how viral replication may be facilitated, whenever degradation of viral proteins by the proteasome is inhibited. The latter can be considered as part of the cellular defense against invading pathogens and inhibition of proteasome activity, as shown here for ONX 0914, might thereby also promote viral replication.

Death of cardiomyocytes by increased viral cytotoxicity during the acute phase of myocarditis in ONX 0914-treated NMRI mice is reflected by higher release of TnT and the cellular injury in the heart triggers an elevated infiltration of inflammatory monocytes, compared to the vehicle group. This adverse inflammatory response is in contrast to our previous demonstration of ONX 0914-mediated reversal of immunopathology in A/J mice. Here, ONX 0914 blocks the detrimental cytokine overproduction, reminiscent of sepsis, and thereby protects from viral pathology and infiltration of myeloid immune cells into the heart [38]. Interestingly, although in NMRI mice the ONX 0914-mediated anti-inflammatory effects are overwhelmed by elevated viral cytotoxicity, a closer assessment of cardiac function indeed confirms some protective properties of ONX 0914 in NMRI mice as well. The decreased end-diastolic volume mice have during the acute state of myocarditis reflects the vascular dilatation mediated by pro-inflammatory molecules within the systemic inflammatory response syndrome. Similar to A/J mice, ONX 0914 reduces this drop of left ventricular filling during infection and thereby partially compensates for the reduced cardiac output during viral myocarditis. Despite preserved ventricular filling, the cardiac output is also reduced in myocarditis in the ONX 0914 group and this is mostly attributed to elevated viral cytotoxicity.

Another aspect, how particularly unselective proteasome inhibitors might affect cardiac function, involves the central role of the proteasome complex for the timely degradation of sarcomeric proteins, the main components for contractile function of cardiomyocytes. Therefore, we also need to consider the selectivity of ONX 0914, an irreversible inhibitor of the i-proteasome [20], for the cardiac proteasome complex. In naive splenic tissue, with a high abundance of residential immune cells and likewise a high constitutive level of i-proteasome expression, ONX 0914 specifically inhibited the i-proteasome, yet over time the efficacy of the respective binding to the i-proteasome declined. Moreover, prolonged ONX 0914 treatment might also result in a shift from i-proteasome selectivity towards elevated inhibition of the standard proteasome as shown here for heart tissue and as previously suggested by others [22]. In fact, since ONX 0914, a covalently reacting and irreversible LMP7 inhibitor, reacts to some degree with β5 [20], it was expected to show diminishing isoform selectivity over time [22]. Our observation, suggesting that de novo formed i-proteasomes are less efficiently inhibited by ONX 0914 over time, might have clinical implications, whenever long-term treatment may be required. In such cases, putative side effects might arise from disturbed proteostasis [10,11], leading not only to the death of immune [57], but also of non-immune cells. Recently, the Nathan group proposed a way to circumvent the issue observed here by introducing N,C-capped dipeptidomimetics, showing a much higher selectivity for LMP7 over β5 [58] than that accomplished by ONX 0914 [20]. Stable selectivity of dipeptidomimetic i-proteasome inhibitors was shown in in vitro studies, however, data on subunit selectivity are not yet available for in vivo experiments. In any case, we see the advantages potentially achievable by irreversible i-proteasome selective inhibitors, since a loss of i-proteasome subunit selectivity and the resulting inhibition of the standard proteasome, as shown here for repetitive treatment with ONX 0914 in the heart, can perturb protein homeostasis, e.g., in tissues with highly abundant standard proteasome expression levels. 

Achievement of partially overlapping inhibition of the standard proteasome subunit β5 by ONX 0914 in cardiac tissue, as shown here, relates back to the observed cardiotoxicity, induced by marketed proteasome inhibitors for the treatment of multiple myeloma [59,60] by their inhibition of the cardiac proteasome complex. Like other cells, cardiomyocytes depend on cellular responses for the timely removal of misfolded proteins and this is orchestrated in part by the UPS. Carfilzomib, a proteasome inhibitor used in the clinics, mainly targets the β5/β5i subunits and causes cardiomyocyte toxicity, most likely through proteotoxic stress resulting from the inhibition of standard proteasome-dependent sarcomeric protein turnover, with subsequent activation of the unfolded protein response and apoptosis [61]. Accumulating dysfunctional sarcomeric proteins can perturb the functional integrity of the sarcomere, required for the contractile function of the heart. Hence, defective activity of the cardiac proteasome, as gradually accomplished by ONX 0914 in acute myocarditis of NMRI mice, might have detrimental effects on cardiomyocyte function. In addition to elevated viral cytotoxicity, the cumulative inhibition of the standard proteasome by an irreversible immunoproteasome inhibitor with general binding capacity for β5 presents a risk of toxicity [13] and might partially explain why NMRI mice showed impaired systolic function upon treatment with ONX 0914, assuming a parallel between this treatment and the effect accomplished by licensed proteasome inhibitors. 

## 4. Materials and Methods

### 4.1. Mouse Studies

Outbred NMRI mice were purchased from Charles River, Germany and kept in the animal facility at Charité University Medical Center, Berlin. Myocarditis was induced in 6-week old male mice by intraperitoneal infection with 5 × 10^5^ plaque forming units (pfu) CVB3 31-1-93 [45]. For the therapeutic i-proteasome inhibitor treatment, mice were injected subcutaneously with 5–10 mg/kg bodyweight (BW) ONX 0914 or the vehicle Captisol daily from day 3 to 8 (acute viral myocarditis). For experiments investigating chronic viral myocarditis, treatments were continued three times per week until day 28. For the prophylactic approach, the general treatment plan remained the same, but injections were instead given starting one day prior to infection. On the final day of each experiment, after echocardiography, organs were collected for further analysis and immediately frozen in liquid nitrogen. Organs were stored at −80 °C. Whole blood was centrifuged at 4 °C and 10,000 rcf for 15 min to separate serum, which was then collected and stored at −80 °C. All animal experiments were in accordance with the local regulations and guidelines as well as approved by the Committee on the Ethics of Animal Experiments of Berlin state authorities (TVA Nr. G103/18 and G0274/13). All efforts were made to minimize suffering.

### 4.2. Echocardiography

Transthoracic echocardiography was performed on a VisualSonics Vevo770 High-Frequency imaging system (FUJIFILM VisualSonics, Toronto, ON, Canada) using a high-resolution scan head CRMV-707B; 15-45Hz). Mice were anaesthetized with 1–2.5% isoflurane and fixed on a heated pad with integrated electrodes for continuous ECG measurement. An experienced technician, who was not aware of treatment allocation, performed all measurements. Standard planes and measurements were obtained using standard M-mode. Functional and dimensional calculations are based on the parasternal long axis view of the left ventricle.

### 4.3. Proteasome Inhibitor ONX 0914 

For subcutaneous administration, ONX 0914 (Cayman Chemicals, Ann Arbor, US.A.) was dissolved at a concentration of 1–2 mg/mL in an aqueous solution of 1 mg/mL Captisol (Ligand Pharmaceuticals, San Diego, CA; USA.) and sodium citrate at pH 3.5. Aliquots of both ONX 0914 and Captisol were stored at −20 °C.

### 4.4. Quantification of Infectious Viral Particles

Organ virus titers were determined by standard plaque assay on HeLa cells grown in monolayers. Experiments were performed in duplicates on 24-well cell culture plates using 10-fold titrations of homogenized organs to infect cells for 30 min at 37 °C. Supernatant was carefully discarded before applying agar overlays (containing MEM, 1% penicillin/streptomycin, 2.3 g/L NaHCO_3_, 9.2% FBS and 0.7% Difco Agar Noble (BD Bioscience, Heidelberg, Germany)). After incubation for 48 h, cells were stained with MTT (3-(4,5-dimethylthiazol-2-yl)-2,5-diphenyl tetrazolium bromide), incubated for at least 1 h and plaques were subsequently counted.

### 4.5. Histology

All histological samples were fixed in 4% formaldehyde overnight before analysis. As previously described, 5 µm thick slices of heart tissue were stained with hematoxylin/eosin and Masson’s trichrome [45].

### 4.6. Flow Cytometry

To generate samples for flow cytometry, hearts were flushed with 10 mL PBS before removal and washed again in PBS before a piece for FACS analysis was weighed and stored in washing buffer (RPMI 1640 with 2% FBS, 1% penicillin/streptomycin, and 30 mM HEPES). Cells were extracted by digestion using 1.0 mg/mL collagenase type 2 (Worthington Biochemical Corporation, Lakewood, NJ, USA) and 0.15 mg/mL DNase I (Sigma-Aldrich, St. Louis, MO, USA.) for 30 min at 37 °C under continuous agitation. EDTA was added at a concentration of 10 mM before samples were processed through a 70 µm cell strainer. Erythrocyte lysis was performed for 3–5 min at room temperature using 10 mM KHCO_3_, 155 mM NH_4_Cl, and 0.1 mM EDTA. Cells corresponding to 20 mg heart tissue were incubated with Fc blocking reagent (Miltenyi) at a concentration of 1:50 for 15 min at 4 °C. Cells were then stained with the antibodies listed below for 15 min at 4 °C. Samples were washed with FACS buffer (PBS with 2% FBS and 2 mM EDTA) and PBS before Fixable Viability Dye eFluor 780 (eBioscience, San Diego, CA, USA.) was applied according to manufacturer’s instructions. Samples were fixed using 2% formaldehyde in PBS for 30 min at room temperature and washed with PBS prior to addition of 123count eBeads (eBioscience) for the quantification of total cell numbers. Antibodies were purchased from BD (CD69 clone H1.2F3, CD4 clone RM4-5, CD8α clone 53-6.7, CD45 clone 30-F11, B220 clone RA3-6B2, CD3 clone 145-2C11), Biolegend (San Diego, CA, USA.) (F4/80 clone BM8, CD11b clone M1/70, Ly6G clone 1A8, CD11c clone N418, Ly6C clone HK1.4) or eBioscience (CD49b clone DX5). Flow-cytometric analysis was conducted on a BD FACSymphony flow cytometer and data were analyzed using FlowJo V10 software (Ashland, Wilmington, DE, USA), applying the gating strategy depicted in Appendix A.

### 4.7. RNA Isolation, cDNA Synthesis, and qPCR

For RNA isolation, Trizol Reagent (ThermoFisherScientific, Waltham, MA, USA) was used according to the manufacturer’s instructions. Samples were incubated with 0.1 U/µL DNAseI (ThermoFisherScientific, Waltham, WA, USA) prior to cDNA synthesis using random hexamer primers and MLV-reverse transcriptase (Promega, Madison, WI, USA). Using TaqMan Fast Universal PCR Master Mix (ThermoFisherScientific, Waltham, MA, USA) as well as primers and probes of TaqMan gene expression assays (ThermoFisherScientific, Waltham, MA, USA), qPCR was performed on a StepOnePlus real-time PCR System ((ThermoFisherScientific, Waltham, MA, USA). CVB3 sequences for primer/probe-combinations used were 5′-CCCTGAATGCGGCTAATCC-3′ (sense), 5′-ATTGTCACCATAAGCAGCCA-3′ (anti-sense) for the primers and 5′-FAM-TGCAGCGGAACCG-MGB-3′ for the probe. The housekeeping control employed was mHPRT, with the sequences 5′-ATCATTATGCCGAGGATTTGGAA-3′ (sense), 5′-ATTGTCACCATAAGCAGCCA-3′ (anti-sense) for the primers and 5′-FAM-TGGACAGGACTGAAAGACTTGCTCGAGATG-MGB-3′ for the probe.

### 4.8. Western Blot Analysis

Tissue samples were homogenized in lysis buffer containing 50 mM Tris-HCl (pH 8.0), 150 mM NaCl, 1% NP40, 1 mM EDTA, 0.1% SDS, 0.5% Na-deoxycholate, 1 × cOmplete protease inhibitor cocktail (Sigma-Aldrich, St. Louis, MO, USA) and 5 µM NEM. Bradford assays were performed to determine protein concentrations (measured on a Synergy HT plate reader (BIO-TEK, Winooski, VT, USA) before samples were diluted to 1 µg/µL in Laemmli buffer. For gel electrophoresis, samples were applied to a 12% acrylamide-bisacrylamide gel and run in an Mini PROTEAN Tetra system (BioRAD, Hercules, CA, USA) before tank blotting was performed using the same system. The following primary antibodies were used for immunostaining: β1 (K43, lab stock), β2 (Abcam, Cambridge, UK), β5 (Abcam, Cambridge, UK), LMP2 (Abcam, Cambridge, UK), MECL-1 (K65, lab stock), LMP7 (K63, lab stock), β-actin (C4, Merck Millipore, Burlington, VT, USA). Secondary IRD680CW or IRDye800CW labeled antibodies (Li-Cor Biosciences, Lincoln, MA, USA) were used for the detection on an Odyssey CLx imager (Li-Cor Bioscience, Lincoln, MA, USA). To control for equal loading of spleen tissue lysates, REVERT® Total Protein stain (Li-Cor Bioscience, Lincoln, MA, USA) was performed according to manufacturer’s instructions. Protein bands were quantified using ImageStudio Light 5.2 (Li-Cor Bioscience, Lincoln, MA, USA) and were normalized to either β-actin or total protein stain. Some blots had a poor signal-to-noise ratio and densitometric analysis was not possible.

### 4.9. Statistics

Data was analyzed using GraphPad Prism 7.0 (GraphPad Software, San Diego, CA, USA). Logarithmic data such as virus titers was transformed logarithmically before data was plotted and statistically analyzed. All data was tested for normal distribution using the D’Agostino-Pearson normality test. For normally distributed data, paired or unpaired *t*-tests were performed. Non-normally distributed data were analyzed using either Wilcoxon-signed rank test or Mann–Whitney test. To compare treatment groups or time points, repeated measurements two-way ANOVA with Sidak’s multiple comparisons were performed. Data is displayed as mean ± SEM unless otherwise indicated.

## Figures and Tables

**Figure 1 cells-09-01093-f001:**
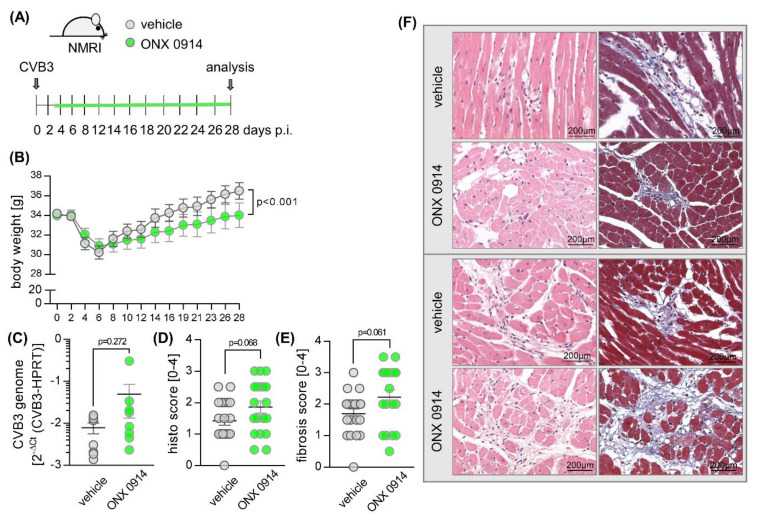
Effect of ONX 0914 on the manifestation of chronic viral myocarditis in NMRI mice. (**A**) Mice were inoculated with 5 × 10^5^ pfu of Coxsackievirus B3 strain 31-1-93. Between days 3 and 8, mice were treated with ONX 0914 (5–10 mg/kg) or vehicle (Captisol) daily—thereafter treatment continued three times per week until day 28 (vehicle *n* = 20; ONX 0914 *n* = 20). (**B**) Body weight was monitored as indicated and analyzed using two-way ANOVA. (**C**) Mice were sacrificed after 28 days. RNA was extracted from heart tissue and Coxsackievirus B3 (CVB3) genome expression was determined by quantitative PCR. Paraffin-embedded, hematoxylin-eosin (**D**) and Masson’s trichrome-stained cardiac tissue (**E**) was taken for histological scoring of cardiac injury and inflammation. (**F**) Two representative images (upper panel from the group treated with 5–7 mg/kg bodyweight ONX 0914; lower panel from the group treated with 10 mg/kg bodyweight ONX 0914) are shown for each treatment group. Data are mean ± SEM and analyzed by unpaired *t*-tests. *p*-values are indicated in each graph.

**Figure 2 cells-09-01093-f002:**
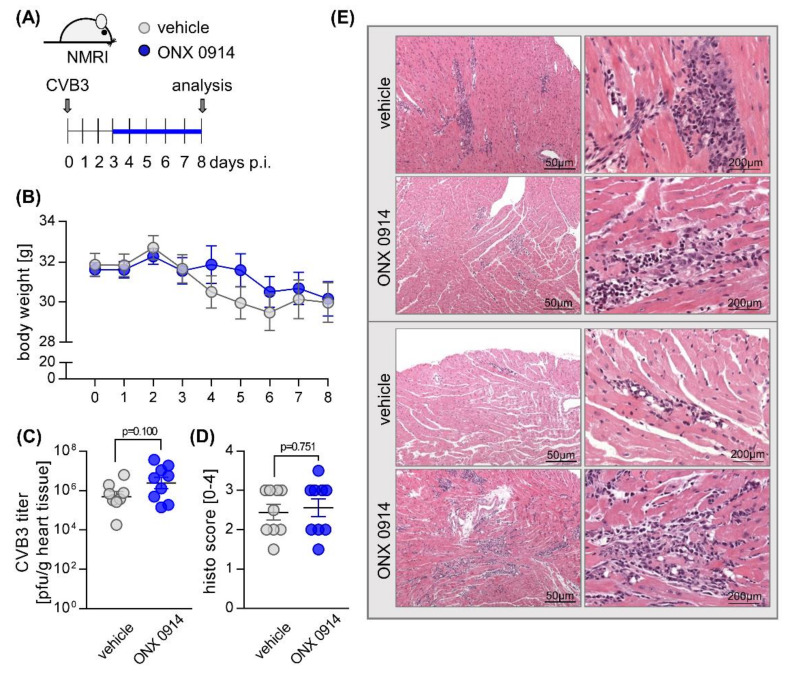
Effect of ONX 0914 on the manifestation of acute viral myocarditis in NMRI mice. (**A**) Mice were infected with 5 × 10^5^ pfu CVB3 strain 31-1-93 (day 0). From day 3, mice received either ONX 0914 or vehicle daily until day 8. (**B**) Body weight was monitored as indicated. (**C**) Viral titers in hearts of both vehicle and ONX 0914-treated mice were determined by plaque assay. (**D**) Mice were then sacrificed, and heart tissue was analyzed microscopically for inflammation and fibrosis. (**E**) Histological images of two representative animals per group are shown. Data are summarized as mean ± SEM. Body weight was analyzed using two-way ANOVA, all remaining data were analyzed by unpaired *t*-tests.

**Figure 3 cells-09-01093-f003:**
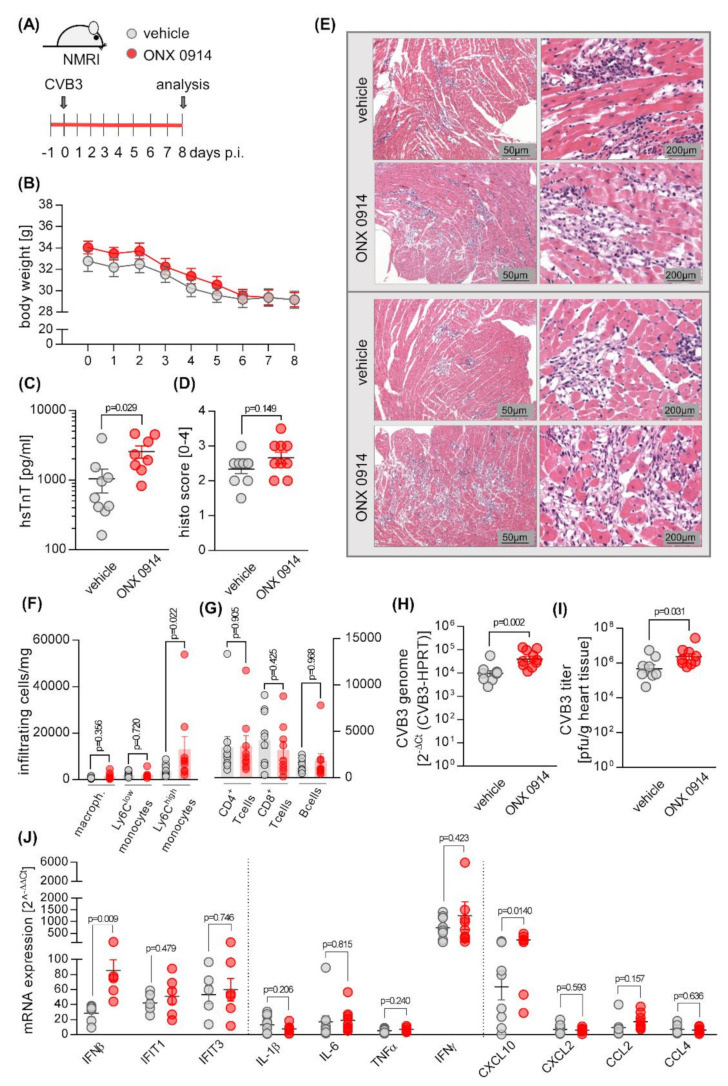
ONX 0914 treatment prior to infection with CVB3 exacerbates virus-mediated heart tissue injury. (**A**) Mice received either ONX 0914 (10 mg/kg bodyweight) or vehicle daily until day 8, starting one day prior to infection with 5 × 10^5^ pfu of CVB3 strain 31-1-93. (**B**) Body weight was monitored daily and analyzed using two-way ANOVA. (**C**) On day 8, blood was drawn for measurement of hsTnT. (**D**) Mice were sacrificed on day 8. For histological analysis, HE-stained heart tissue sections were scored microscopically for myocardial cell death and inflammation. (**E**) Images from two representative mice per treatment group are shown. (**F**,**G**) Single cell solutions from heart tissue were generated and analyzed by flow cytometry for infiltration of immune cells. The viral load in infected mouse hearts was determined by quantification of the CVB3 genome, applying qPCR (**H**) and by quantification of viral titers, using plaque assay (**I**). Heart tissue was homogenized and RNA was extracted for qPCR-analysis of the indicated genes (**J**). Relative mRNA levels were normalized to the housekeeping gene HPRT and mRNA induction was normalized to data obtained with non-infected mice. Data are mean ± SEM, *p*-values are indicated in each graph. If not indicated otherwise, data was analyzed using unpaired *t*-test.

**Figure 4 cells-09-01093-f004:**
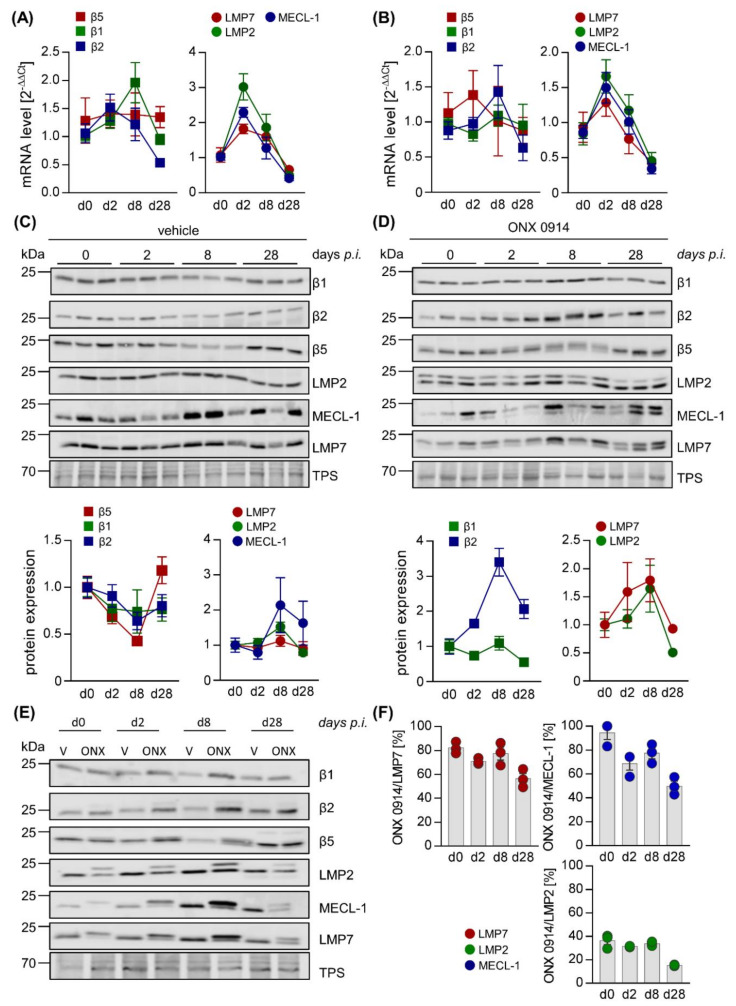
ONX 0914 selectively binds to i-proteasome subunits in the spleen during acute and chronic stages of viral myocarditis. Total RNA was extracted from spleen tissue of vehicle- (**A**) and ONX 0914- (**B**) treated mice sacrificed at different time points after CVB3 infection and mRNA expression levels were determined by qPCR for subunits β5, β1, and β2 (standard proteasome) as well as their i-proteasome counterparts LMP7, LMP2, and MECL-1, respectively. (**C**,**D**) Protein homogenates were obtained and subjected to Western blot analysis, showing the protein expression for the indicated catalytic subunits of the proteasome complex for three different animals per group (vehicle: C; ONX 0914: D) on days 0, 2, 8, and 28 post-infection. Analysis of β5 and MECL-1 did not yield reliable densitometric readings and their protein expression is subsequently not shown. The section of the blot corresponding to 45 kDa treated with total protein stain indicates protein loading in each lane. Protein expression (mean ± SEM) was quantified using Image Studio Lite. Relative protein expression levels normalized to baseline controls and protein loading are shown. (**E**) To compare efficacy of i-proteasome inhibition at the different stages of infection, protein homogenates, obtained from a representative animal of each group (vehicle- and ONX 0914 treatment, days 0, 2, 8, and 28), were loaded onto the same SDS gel and Western blot analysis was performed as indicated. Total protein stain served as the loading control. (**F**) Covalent binding of ONX 0914 to the catalytic subunits of the proteasome induces an upward shift of the protein band for the respective proteasome subunit, indicative of an elevation of its molecular weight by the binding of ONX 0914. Based on the shifts detected with the Western blot analysis depicted in (**C**) and (**D**), the relative inhibition achieved by ONX 0914 was calculated for each i-proteasome subunit. This was accomplished through division of the signal from the upper (ONX 0914-bound) band by the total expression signal, which was calculated by the addition of both upper and lower bands in each individual lane.

**Figure 5 cells-09-01093-f005:**
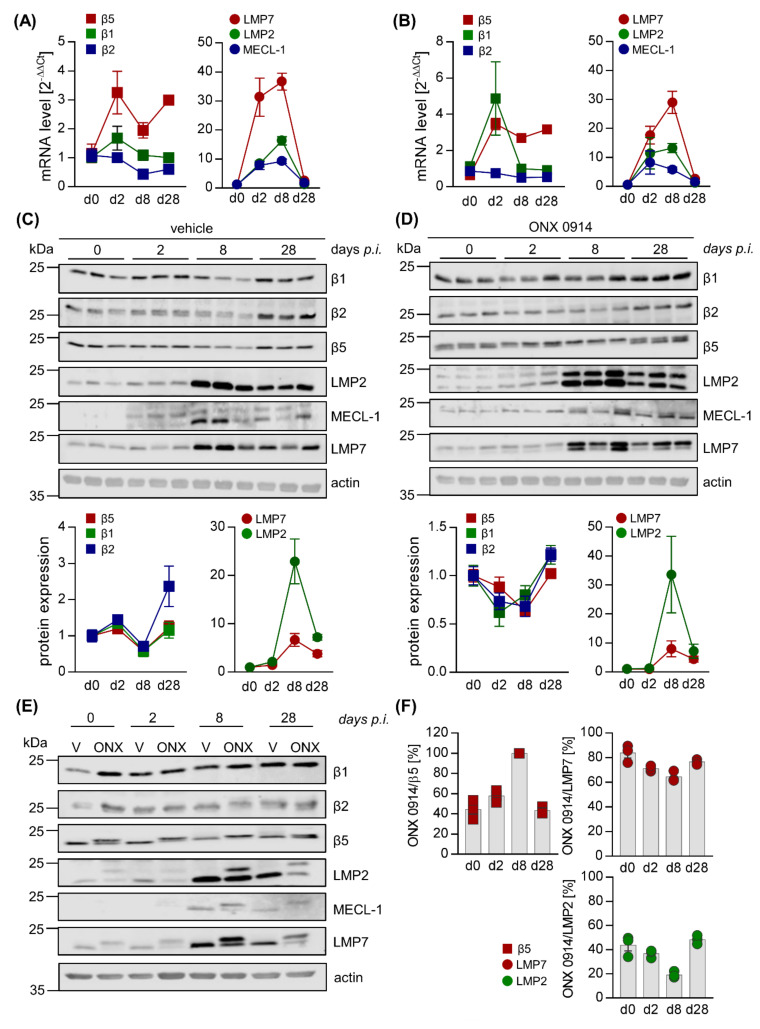
Loss of selectivity of ONX 0914 for the i-proteasome during acute viral myocarditis in heart tissue. Total mRNA was extracted from heart tissue of vehicle– (**A**) and ONX 0914– (**B**) treated mice sacrificed at different time points post CVB3 infection and expression levels were determined by qPCR for subunits β5, β1, and β2 (standard proteasome) as well as their i-proteasome counterparts LMP7, LMP2, and MECL-1, respectively. (**C**,**D**) Protein homogenates were obtained and subjected to Western blot analysis, showing the protein expression of the respective subunits for three different animals per group (vehicle: C; ONX 0914: D) on days 0, 2, 8, and 28 post-infection for the indicated catalytic subunits of the proteasome complex. Actin indicates protein loading. Protein expression (mean ± SEM) was quantified using Image Studio Lite. Relative protein expression levels normalized to baseline controls and protein loading are shown. We were not able to quantify the protein expression levels for MECL-1. (**E**) To compare efficacy of i-proteasome inhibition at the different stages of infection, protein homogenates obtained from a representative animal for each group (vehicle- and ONX 0914 treatment, days 0, 2, 8, and 28) were loaded onto the same SDS gel and Western blot analysis was performed as indicated. Actin served as loading control. (**F**) Based on the shifts of the molecular weight of proteasome subunits detected by Western blot analysis (**C**,**D**), the relative inhibition achieved by ONX 0914 was calculated for each i-proteasome subunit, as well as for β5 of the standard proteasome (as described in Figure 4F).

**Table 1 cells-09-01093-t001:** Analysis of cardiac function in NMRI mice during chronic myocarditis.

Collumn Name	Vehicle		ONX 0914	
	Baseline	Day 28	Baseline	Day 28
**Heart rate (bpm)**	417 ± 18	440 ± 18	440 ± 18	426 ± 14
**Trace EF (%)**	46 ± 2.5	48.1 ± 2.9	43.6 ± 2.6	42.7 ± 2.0
**Cardiac output (mL/min)**	13.6 ± 1.0	14.6 ± 1.0	14.3 ± 1.1	14.0 ± 0.8
**Stroke volume (µL)**	32.4 ± 1.8	32.7 ± 1.5	32.0 ± 1.9	33.0 ± 1.8
**Vol d (µL)**	70.5 ± 1.7	70.0 ± 3.1	74.2 ± 3	77.0 ± 2.2
**Vol s (µL)**	38.1 ± 1.9	37.3 ± 3.0	42.2 ± 2.9	44.2 ± 2.0
**LVID-d (mm)**	4.3 ± 0.1	4.3 ± 0.1	4.3 ± 0.1	4.5 ± 0.1
**LVID-s (mm)**	3.2 ± 0.1	3.1 ± 0.1	3.2 ± 0.1	3.5 ± 0.1

Echocardiography was performed prior to infection (day 0) and on day 28 post-infection. After infection, mice received ONX 0914, starting on day 3 (*n* = 20 ONX 0914). Age- and gender-matched vehicle-treated mice served as controls (*n* = 20 vehicle). Data shown are mean values ± SEM and were analyzed using repeated measurements two-way ANOVA, yielding no significant changes. EF = ejection fraction; bpm = beats per minute; Vol d/s = end-diastolic/-systolic left ventricular volume; LVID-d/s = left ventricular diameter at diastole/systole.

**Table 2 cells-09-01093-t002:** Analysis of cardiac function in NMRI mice during acute myocarditis.

	Vehicle		ONX 0914	
	Baseline	Day 8	Baseline	Day 8
**Heart rate (bpm)**	427 ± 16	393 ± 16	500 ± 13 *	405 ± 23 ^§^
**Trace EF (%)**	50.7 ± 1.7	48.4 ± 2.6	57.4 ± 2.3	49.1 ± 3.2 ^§^
**Cardiac output (mL/min)**	14.4 ± 1.2	10.0 ± 0.9 ^§^	18.6 ± 1.2 *	12.1 ± 1.3 ^§^
**Stroke volume (µL)**	33.6 ± 2.1	25.2 ± 1.4 ^§^	37.3 ± 2.4	29.6 ± 2.3 ^§^
**Vol d (µL)**	66.5 ± 4.0	52.8 ± 3.0 ^§^	65.5 ± 4.0	60.3 ± 2.6
**Vol s (µL)**	33.0 ± 2.4	27.6 ± 2.6	28.1 ± 2.5	30.7 ± 2.3
**LVID-d (mm)**	4.2 ± 0.1	4.0 ± 0.1	4.2 ± 0.1	4.2 ± 0.1
**LVID-s (mm)**	3.1 ± 0.1	3.0 ± 0.1	2.9 ± 0.1	3.2 ± 0.1

ONX 0914 (10 mg/kg BW) was initiated in naive NMRI mice one day prior to infection. Echocardiography was performed at baseline (day 0) and on day 8 post-infection. Age- and gender-matched vehicle-treated littermates served as controls (*n* = 10 vehicle and *n* = 10 ONX 0914). Data shown are mean values ± SEM and were analyzed using repeated measurements two-way ANOVA, followed by Sidak’s multiple comparison test. § indicates significant differences in the respective treatment group at day 8 compared to this groups’ baseline measurement. * indicates significant differences regarding ONX 0914 treatment prior to infection. EF = ejection fraction; bpm = beats per minute; Vol d/s = end-diastolic/-systolic left ventricular volume; LVID-d/s = left ventricular inner dimension at diastole/systole.

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
