# Peer review of "ONX 0914 Lacks Selectivity for the Cardiac Immunoproteasome in CoxsackievirusB3 Myocarditis of NMRI Mice and Promotes Virus-Mediated Tissue Damage"

_cells, 2020, doi:10.3390/cells9051093_

Round 1
Reviewer 1 Report
The paper from the group of Beling reports on the effects of putative selective inhibitor ONX 0914 on the CoxsackievirusB3 infection in NMRI mice (this results in the development of acute and chronic myocarditis). The results obtained show that ONX 0914 actually worsens the cardiac issues due to unselective perturbation of cardiac proteasome function.
I find the study very useful for the development of the 'immunoproteasome inhibition field' and do recommend publication in Cells. There are, however, some minor issues that need to be addressed to make the paper more wholesome.
- There are several new review papers on the development of i-proteasome selective inhibitors (of both patent and non-patent literature) that were published after the paper by Huber and Groll from 2012. These should be added next to reference 11 on page 2, line 57, in order to keep the readers up-to-date.
- Page 2, line 64. It is true that ONX 0914 is the most studied 'i-proteasome selective compound' but it should be stated that there are more selective beta5i inhibitors (and of other subunits as well) available. I acknowledge the authors' realization regarding this issue (on page 15), but I would prefer that this is also briefly mentioned in the Introduction.
- Related to point 2: For future studies, I would highly recommend to the authors to refrain from using solely ONX 0914 in the research, because, as the authors mention several times, ONX 0914 is not very i-proteasome selective. Therefore, more selective i-proteasome inhibitors should be used, such as DPLG-3, KZR-616 and M3258 (LPM-7-IN-1). The latter two are commercially available, whereas DPLG-3 (supposedly the most selective i-proteasome inhibitor, which is also a non-covalent inhibitor as oppose to ONX 0914 and KZR 616) can be readily synthesized as reported by the original authors (reference 23) and other groups working in the field of i-proteasome inhibitors (e.g. Shannon Schiffrer, E. et al. Med Chem Commun 2019).
- It should also be noted on Page 2, line 70, that in reference 23, other i-proteasome selective inhibitor was the focus, i.e. DPLG-3.
- Did the authors check the specificity of the antibodies for individual constititive proteasome and i-proteasome subunits that were used in this study? Given my experience this might be an issue.
Reviewer 2 Report
In the study were evaluated the effects of immunoproteasome inhibitor ONX 0914 on development of CoxsackievirusB3-induced myocarditis in NMRI mice. I have some questions and comments:
1. You wrote (lines 208-211): “On the other hand and in line with the elevated myocardial cytotoxicity in ONX 0914-treated mice, we found a reduced left ventricular ejection fraction in this group, illustrating systolic dysfunction as a main effector of the reduced cardiac output. Vehicle-treated NMRI mice had no relevant decline of the ejection fraction at the acute state of myocarditis.”
In this case is necessary to mention that the ejection fraction in ONX 0914-treated mice at the day 8 was similar (or even slightly increased) to the ejection fraction at the acute state of myocarditis in vehicle-treated animals.
How you explain the increased baseline values of heart rate and EF in ONX 0914-treated mice? Which mechanisms participate in these effects of ONX 0914?
2. I have question and comments to data presented in Figure 5C and 5D.
Relative protein expression was expressed as ratio of specific protein (proteasome, i-proteasome subunit) to actin? If yes, based on the presented Western blot records, should not be values of relative expression for LMP2 and LMP7 at day 8 or day 28 over 1.0? Intensity of reaction for both LMP2 and LMP7 seems to be higher than intensity of reaction for actin. You presented graphs with values in range 0.2-0.4.
3. The data presented in graps not always correspond with analysis records.
Is protein expression of LMP2 or LMP7 in ONX 0914-treated animals at the day 28 really the same or lower than at the day 0? See data presented in western blot record and graph in Figure 4D.
4. Legend to the Figure 4. There is sentence: “Based on the shifts detected with the Western blot analysis depicted in (C) and (D), the relative inhibition achieved by ONX 0914 was calculated for each i-proteasome subunit. This was accomplished through division of the signal from the upper (ONX 0914-…”
This sentence in manuscript is not complete. Should be there “through division of the signal from the upper (ONX 0914-subunit) band to the lower (subunit) band”?
5. In Legend to the Figure 4E you wrote: “Actin served as loading control”. Is it correct? In this Figure 4E you show record of total protein stain (TPS).
Reviewer 3 Report
This study is designed to investigate the effects of ONX 0914, an immunoproteasome-selective inhibitor, on the pathogenesis of coxsackievirus B3 (CVB3) myocarditis. The investigators have previously studied this in C57BL/6 and A/J mice (Althof et al. 2018), and they now use a different mouse strain, outbred NMRI mice, in an effort to better characterize effects of ONX 0914 treatment on chronic viral myocarditis. The overall conclusions of the study suggest that ONX 0914 had minimal short-term or long-term effect on viral replication, cardiac inflammation, cardiac fibrosis, or cardiac function when treatment is started at 3 days post infection. In contrast, beginning treatment before infection impaired control of viral replication in the heart during acute infection, increased some aspects of virus-induced cardiac inflammation and cardiac myocyte damage, and worsened some (but not all) echocardiographic measurements of cardiac function.
The investigators use a variety of complementary approaches to define these effects. It is particularly helpful to see the combination of microbiology, immunology, and physiology used to characterize effects in the mouse model. Some of the findings during acute infection differ from those that the group presented in a previous paper (Althof et al. 2018). In that report, ONX 0914 treatment increased viral replication and decreased IFN-beta production in ONX-treated C57BL/6 mice. In A/J mice, ONX treatment had no substantial effect on viral replication but increased survival, decreased weight loss, decreased inflammation, and decreased the drop in cardiac output seen in infected animals. Collectively, these findings raise some interesting points about mouse strain-based differences in viral pathogenesis and immunoproteasome activity, although it would be helpful if these were addressed in more detail in the discussion section.
Efforts are made to directly quantify the effects of ONX 0914 on proteasome and immunoproteasome subunit activity using Western blots. The results of those assays seem to indicate that ONX 0914 is acting in a way that is not specific to the beta 5i immunoproteasome subunit, which would make it somewhat difficult to ascribe any observed effects to immunoproteasome inhibition. As noted below, some aspects of these assays should be clarified to aid in their interpretation.
In general, this study adds some information to the overall understanding of the role played by the immunoproteasome in CVB3 myocarditis. There are many aspects of the work, detailed below, that could be addressed in order to improve the impact and overall appeal of the study.
Major Points:
1) Mice were treated with ONX 0914 at 5-10 mg/kg given daily. It is unclear why a range of doses was used instead of a single dose throughout an experiment. It seems possible that this could contribute to some variability in effect from mouse to mouse and from one experiment to another, if some mice received twice as much drug as others. This would seem particularly relevant for comparisons to effects in other mouse strains in the group’s previous publication (Althof et al. 2018), in which 10 mg/kg/dose was used in all cases.
2) Densitometry results from Western blots to detect proteasome and immunoproteasome subunit proteins are used to assess inhibitory effect of ONX 0914, with ratios of larger bands (presumably subunit bound to ONX 0914) to smaller bands (unbound subunit) serving as a measure of inhibition. This could be problematic in the setting of increasing immunoproteasome subunit expression (for example, at 8 days post infection in the heart) with a constant dose of ONX 0914, where decreases in the ratio of bound to unbound protein (as in Figure 5F for LMP2 and LMP7) may simply be an effect of insufficient ONX 0914 rather than altered selectivity of the drug. It is also strange that the size discrepancy between the two bands for various subunits seems to vary between the subunits being assessed. Because the gel shift would be caused by binding of the same factor (ONX 0914), a similar size discrepancy would be expected. The presence of a similar pattern of double bands for MECL-1 in vehicle-treated hearts, although not quantified (Figure 5C), raises additional questions about the specificity of this approach. In general, direct assays of immunoproteasome subunit activity would be useful additions in order to better assess effects of infection and effects of ONX 0914.
3) The same densitometry approach is used to assess effects of ONX 0914 on activity of the beta 2 standard proteasome subunit. Although there does appear to be a shift in band size for beta 5 in ONX 0914-treated samples, the difference is very small compared to differences seen for immunoproteasome subunits, often seeming to blend together into a broad band. Again, direct assays of subunit activity would seem better suited to define any nonspecific effects of ONX 0914 on the beta 5 subunit.
4) In Figures 4 and 5, quantified data for subunit mRNA and protein levels are presented separately for each group (vehicle, ONX 0914). Comparisons between the two groups would seem relevant in order to better assess inhibitory effects of ONX 0914. Consideration could be given to presenting data from vehicle and ONX 0914 groups for a given subunit together in the same graph to facilitate those comparisons.
5) There are several places in which differences that are not statistically significant are described by the authors in the text in ways that make those differences sound more prominent. Examples include line 131 (“enhanced viral RNA copy number”), line 144 (“mildly elevated scores”), and line 169 (“slightly increased CV3B titer”), lines 173-174 (“evidence of a mild exacerbation of cardiac inflammation in the ONX 0914 group”), lines 184-185 (“evidence of increased myocardial injury”). Care should be taken to describe these findings accurately, taking the results of statistical testing into account.
6) The authors indicate that ONX 0914 treatment did not affect cardiac fibrosis following infection. This assessment appears to be based on histological evaluation of hematoxylin and eosin-stained sections in Figure 1. Fibrosis is not clearly present in the images included in the figures. Other staining techniques would likely be better to evaluate fibrosis. One such technique, Mason’s trichrome staining, is described in the methods section but not included in the results or figures. That type of histologic evaluation and/or quantification of fibrosis in some way would help to solidify any conclusions about effects of immunoproteasome inhibition on fibrosis.
7) Inclusion of physiologic data (echocardiography in Tables 1 and 2) is very nice to see. However, statistical analyses performed on the echocardiography data in Table 2 should be clarified. The asterisk is noted to indicate differences compared to ONX 0914 treatment prior to infection, but those are only used for two “baseline” measurements in the ONX 0914 group. It is not clear to me what is being compared to those “baseline” measurements, which evidently are at day 0 (presumably pre-infection). Although it would seem relevant, it is also not clear that comparisons have been made between the two groups at 8 days post infection – in other words, comparing day 8 vehicle to day 8 ONX 0914. Interestingly, there appear to be some differences between baseline (again, presumably pre-infection) measurements in the vehicle and ONX 0914 groups. It would be helpful to address those potential differences, which may confound comparisons between the groups after infection.
8) In in the introductory sentence to section 2.2, the authors state that ONX 0914 treatment during acute myocarditis “has no relevant effect on systemic signs of viral infection, such as weight loss, cardiac output, or survival” (lines 225-228). While ONX 0914 did not abrogate virus-induced decreases in cardiac output (Table 2), it is not clear whether or not it actually worsened the magnitude of that decrease to a significant degree compared to changes seen in vehicle-treated mice. Additional comparisons between vehicle-treated and ONX 0914-treated groups would be interesting to see.
Minor Points:
9) In the abstract, the authors state that “immunoproteasome inhibitors allow for specific targeting of the proteasome in immune cells…” This statement is somewhat confusing, since ONX 0914 has preferential activity for the immunoproteasome beta 5i subunit rather than standard proteasome subunits. In addition, it should have inhibitory activity in other non-immune cell types in which immunoproteasome activity is induced, particularly in the context of infection.
10) In the introduction (lines 90-93), the authors state, “The i-proteasome plays an important, yet not fully explained role in the process of CVB3-triggered cardiac inflammatory immune response, with both knockout and inhibition of the multi-catalytic protease exacerbating the disease in C57BL/6 mice.” However, the two references provided actually seem to provide conflicting results, with LMP7 deficiency exacerbating disease (Opitz et al. PLoS Pathogens 2011) but ONX 0914 treatment preventing severe disease in A/J mice (Althof et al. EMBO Mol Med 2018). This should be clarified.
11) It is unclear why total protein (in this case, a portion of a gel stained for total protein from a region of the gel that is different than regions shown for individual protein staining) is used to demonstrate protein loading in Figure 4, while actin staining is used in Figure 5. Using a similar approach for the two figures would be helpful.
12) Figure 3J demonstrates significant differences between groups in IFN-beta and CXCL10 mRNA levels, but only IFN-beta is addressed in the text of the results section (lines 196-199). Some discussion of any potential relevance of CXCL10 would be helpful.
13) The authors use the term “viral cytotoxicity” in a variety of places, but it is not entirely clear until later in the discussion how this is defined (largely by cardiac myocyte damage as assessed by serum troponin levels). It would help to define the term when it is first used.
14) The authors discuss a perceived decrease in the efficacy of ONX 0914 over time based on data presented in Figure 5, particularly Figure 5F. Reasonable potential explanations for this waning effect are given. However, it should also be noted that the frequency of ONX 0914 dosing decreased from daily to three times per week after 8 days post infection. This could provide a simple explanation for the observation.
15) Lines 200-211 appear to be a paragraph in the results section but are formatted as if they are a figure legend.
16) For Figures 4 and 5, it would be helpful to indicate the organ (spleen or heart) being evaluated in the figures themselves. In the figure legends for those figures, it would also be helpful to describe infection with CV3B. Of note, the figure legend for Figure 5 appears to be missing important text, as it starts with descriptions of 5C and 5D.
Round 2
Reviewer 2 Report
Authors answered questions and comments that I have addressed to them by revision of the previous version of manuscript and included corresponding changes (based also on my comments) to the revised manuscript.
Reviewer 3 Report
The majority of the concerns that I raised in my initial review have been adequately addressed by the authors in this revision.
Further clarification of the scale in Figures 4A-D and 5A-D would be helpful. I had originally suggested that protein data in Figures 4C/D and 5C/D be presented such that scales are similar for vehicle and ONX-treated groups. In other words, the scale for the left-hand graph in 4C (currently 0 to 1.5) would be the same as the scale for the left-hand graph in 4D (currently 0 to 4), and the same for the right-hand graphs in 4C and 4D as well as the corresponding graphs in Figure 5C and 5D. Although the data are certainly interpretable as currently presented, this change would facilitate direct comparisons between vehicle and ONX-treated groups for each standard proteasome or immunoproteasome subunit.
mRNA data are presented in the same way in Figures 4A/B and 5A/B. The scales for those figures were largely comparable in the first version of the manuscript. Unfortunately, some of those axes have now been changed. Again, though, the data are interpretable as currently presented.